# Ethnicity, Social, and Clinical Risk Factors to Tooth Loss among Older Adults in the U.S., NHANES 2011–2018

**DOI:** 10.3390/ijerph19042382

**Published:** 2022-02-18

**Authors:** Haeok Lee, Deogwoon Kim, Andrew Jung, Wonjeong Chae

**Affiliations:** 1Nursing Department, University of Massachusetts Boston, Robert and Donna Manning College of Nursing and Health Sciences, Boston, MA 02125, USA; deogwoon.kim001@umb.edu; 2College of Dentistry, New York University, New York, NY 10012, USA; aj1827@nyu.edu; 3Office of Strategic Planning, Healthcare Policy and Strategy Task Force, Yonsei University Health System, Seoul 03722, Korea; wjchae0816@yuhs.ac

**Keywords:** older adults, oral health, health disparities, ethnicity, dementia, social factors

## Abstract

Background. Many older adults suffer from poor oral health, including tooth loss, and disparities among racial/ethnic and socially disadvantaged populations continue to exist. Methods. Data were obtained from the National Health and Nutrition Examination Survey among the adult population in the U.S. The prevalence of edentulism and multiple regression models were conducted on 15,821 adults, including Asians, Blacks, Hispanics, Whites, and others to assess the relationships between tooth loss and their predictors. Results. The prevalence of complete tooth loss increased with age from 0.7% for ages 20–44 to 20.2% for ages 65 and over. There are disparities in complete tooth loss regarding race/ethnicity, with the highest percentages (9%) among Whites and Blacks and the lowest percentages among Asians (3%) and Hispanics (4%). After adjusting for predictors, their impact on tooth loss was not consistent within racial/ethnic groups, as Asians had more tooth loss from Model 1 (β = −1.974, *p* < 0.0001) to Model 5 (β = −1.1705, *p* < 0.0001). Conclusion. Tooth loss was significantly higher among older adults and racial/ethnic groups even after controlling for other predictors among a nationally representative sample. The findings point to the fact that subgroup-tailored preventions are necessary.

## 1. Introduction

According to the World Population Prospects 2019, by 2050, 1 in 6 people in the world will be over the age of 65, up from 1 in 11 in 2019 [1]. There is an urgent call to respond to major changes in the demographic composition of the world population and health related to rapidly growing aging. Aging is one aspect of successful human history, as improved nutrition, hygiene, and other factors have contributed to increasing life spans worldwide. However, dementia and oral disease are major challenges in caring for older adult populations [2,3,4,5,6]. Most oral diseases result from complex interactions influenced by genetic, biological, socioeconomic and behavioral health factors [7,8,9,10]. Globally, commonly reported poor oral conditions associated with older individuals include tooth loss, dry mouth (xerostomia), and higher numbers of dental caries and periodontal disease [11,12]. Oral diseases are some of the most expensive diseases, accounting for $545 billion in direct and indirect costs globally [13]. In the U.S., preventive dental costs can incur tremendous costs at $60.5 billion of all healthcare spending, which is far greater than health spending on other disorders, including lung cancer, drug use disorders, and alcohol use in 2016 [14]. Studies provide evidence of an association between tooth loss and dementia, which was stronger for older adults [6,15,16]. Thus, the socioeconomic burden of oral disease is expected to increase substantially, as there is an increasing aging population globally.

According to the Centers for Disease Control and Prevention (CDC) report, one in six American older adults at an age of 65 and above suffers from tooth loss and are completely edentulous [17]. The prevalence of complete tooth loss among older adults aged 65 and above was 12.9%, and Black older adults had a higher prevalence of complete tooth loss than White older adults (25.4% vs. 10.9%) [3]. Ethnic differences in missing teeth may be related to multiple factors from the individual, social, and health care system levels [10,18,19,20,21]. However, relative contributions of physical and sociocultural factors to ethnic differences in tooth loss have not been reported. There is scarce or little population-based data to clearly identify the unique oral health problems in rapidly growing racial/ethnic groups—Asian Americans and Hispanics in particular—who have linguistic and cultural barriers to accessing health care in the U.S.

Asians are the fastest growing racial/ethnic group in the U.S., with a population of more than 14 million as of 2010, which is projected to grow to nearly 36 million in 2060 [22]. However, in the CDC’s “Healthy People 2020” report, as in most other national health reports, data of the prevalence of complete tooth loss were not provided on Asian Americans [23]. The most common notations in these reports regarding Asians are: “data have not been analyzed (DNA)”, “data have not been collected (DNC)”, and “data are statistically unreliable (DSU)”. Thus, the lack of accurate population-based data from this ethnic group masks their health needs, because no data clearly attest to their unique health problems. In addition, it offers no defined baseline of the health status and health behavior of Asian Americans from which goals can be set and evaluated. Additionally, the absence of baseline data makes it almost impossible for grant funders, researchers, and practitioners to know where to target and how to reduce the health inequity gap [24]. 

Oral disease is common among older adults and involves tooth loss, and it is imperative to provide baseline data of tooth loss and its correlators reflecting race/ethnicity among adult populations. Therefore, this study is aimed at: (1) examining ethnic differences (Hispanic, non-Hispanic White, non-Hispanic Black, non-Hispanic Asian) in tooth loss among American young adults (20–44 years), middle-aged (45 to 64 years), and older adults (65 years and older); (2) examining the effects of bio-socio-culture-health behavior and health access factors on tooth loss in three different age groups; and (3) examining racial/ethnic differences in tooth loss while controlling important predictors (covaries).

## 2. Materials and Methods

### 2.1. Data Source and Study Population

The study extracted data from a nationally representative sample of adults, the National Health and Nutrition Examination Survey (NHANES). The NHANES was conducted by the National Center for Health Statistics of the CDC [25]. The survey participants were selected based on households, and they provided informed written consent for the survey. In the NHANES study, a household is defined as the first household member 18 years of age or older listed on the household member roster, who owns or rents the residence where members of the household reside. As for oral health measures, this includes the information collected during the home interview, the mobile examination center (MEC) examination, and specimens for lab analyses [26]. The home interview oral health questions covered topics including dental visit frequency, perceived oral health status, the receipt of preventive health information, oral pain, and periodontal disease. Details of survey and clinical oral examination methods have been previously published [27]. Data from the 2011 to 2018 NHANES were used, since NHANES has been oversampling Asian Americans in addition to traditionally oversampled groups, including Hispanic and non-Hispanic Blacks [25]. Though the data are available up to 2020, due to COVID-19, complete sets of survey data are only available until 2018. These analyses were based on de-identified public-use data. A total of 15,161 participants who were over 20 years old were selected for the study from the survey cycle of 2011–2018. Out of the total individuals, 48.7% (*n* = 7386) were males and 51.3% (*n* = 7775) were females. The final analysis was restricted to the dependent variables of tooth loss and nine correlates of demographics, socioeconomics, health risks, oral health status, and health care access that are mostly used in epidemiological studies [8,9,10,28,29].

### 2.2. Measures

#### 2.2.1. Dependent Variable

The number of missing teeth was used as the dependent variable. For the study, we reported it as two forms: categorical (four groups: none (0), 1 to 5 teeth, 6 to 31 teeth, all (32)) and continuous. Partial loss was measured as missing at least one tooth vs. no tooth loss. Complete loss was measured as complete tooth loss vs. none to any number of tooth loss. The variable was created by adding individual missing tooth counts if the individual did not have any missing teeth, and ranged from zero (full set of teeth) to 32 teeth.

#### 2.2.2. Correlated Variables

We selected variables related to tooth loss based on a literature review [8,9,10] and classified them into five domains, including demographic, socioeconomic, health risk, oral health condition, and health care access.

The demographic domain included sex, age, and race/ethnicity. Race/ethnicity was assessed by self-reporting as non-Hispanic White, non-Hispanic Black, Hispanic, non-Hispanic Asian and other. We will hence refer to non-Hispanic White, non-Hispanic Black, and non-Hispanic Asian as Asian, Black, and White, respectively. Age was considered in this study as a major explanatory variable with four categories (20–44; 45–64; and >65) and was included in all analyses.

Socioeconomic domains were education and income. Education included three categories (less than high school, high school, and above high school); and income was in four categories (up to $2099, 2100 to 5399; 5400 to 8399; 8400).

The health risk domain, related to oral health, included diabetes and smoking because of their known relationship with tooth loss [8,10,30]. Diabetes indicated a self-reported diagnosis of diabetes vs. non-diabetic. Smoking was categorized as every day, some days, and not at all.

The oral health domain included self-reported periodontal bone loss and previous gingival disease.

The health care access domain was measured with health care insurance and the last dental care visits. Dental visits included routine check-ups within the past 6 months or not. The health insurance variable was categorized as whether participants had private health coverage or not, since most people 65 years and older have Medicare.

### 2.3. Data Analysis

First, a descriptive data analysis assessed the distribution of all variables for all participants, and by tooth loss (one or more as an interval measure) and total tooth loss. The *t*-tests, ANOVA, and chi-squared tests were employed to investigate the differences in the number of missing teeth and variables. Then, multiple regression was performed to predict tooth loss among the sample, using all variables found to be significant in bivariate analysis, as well as variables with prior significance [8,9,10] using a generalized lineal model (GLM). The first model was adjusted for age, sex, and ethnicity. The second model was adjusted for education and income, and the third model was adjusted for the tooth loss-related risk factors of smoking and diabetes. The fourth model was adjusted for a history of bone loss and gingival disease, as well as self-rated gingival health. The last model was additionally adjusted for private insurance and the last visit to a dentist. For multi-category variables, the reference group was chosen as the category generally associated with the lowest risk of tooth loss. For those selective variables, the variance inflation factors (VIFs) test was conducted to examine the multi-correlation between independent variables that all variables showed less than 10, which indicated no correlation. Results after analyses were presented with a beta-coefficient and adopted a significance level of 0.01 (*p*-value < 0.01 was considered statistically significant). All analyses were performed by SAS version 9.4 (SAS Institute, Cary, NC, USA).

## 3. Results

The characteristics of the 15,161 participants are shown in Table 1. From the study sample of adults older than 20 years of age, 51% were female and 39% were non-Hispanic White, 23% Hispanic, 22% non-Hispanic Black, and 12% non-Hispanic Asian. Overall, 52% were born in the U.S., the majority (82.5%) had health insurance, only 51% had private insurance, and 36.5% reported utilizing dental services within 6 months. Related to health behavior risk factors, 14.6% (*n* = 2218) had diabetes and 11.6% (*n* = 1756) smoked every day. Twenty-two percent reported a history of gum disease, 13.8% a history of bone loss, and 38% reported that their gum health was very good.

The overall mean of missing teeth was 9.1 (SD ± 9.0); the mean score for the age group of 20–44 was 4.1, for 45–64 it was 8.4, and for those aged 65 and higher it was 10.7. Eight percent had complete tooth loss, 45.2% had 1–5 missing teeth, and 41.4% had 6–31 missing. The prevalence of complete tooth loss increased with age from 0.7% for ages 20–44 to 20.2% for ages 65 and over. There were disparities in complete tooth loss in race/ethnicity with a high percentage (9%) among White and Black participants, and the lowest percentages among Asians (3%) and Hispanics (4%). Participants who had low education, low income, health risk factors, a dental condition, no private insurance, and no recent dentist visits were more likely to have fewer teeth compared with their reference groups.

Table 2 separately shows the association of selected predictors and the number of missing teeth for different age groups, along with the results of the Chi-square tests. Comparing the young adult population (aged 22–44 years) and the older adult population (>65 years), the results indicate that members of young adult groups have significantly fewer missing teeth than members of older adult groups. As per the race group, those who did not have missing teeth were in the order of Black (16.3%), Hispanic (12.9%), Asian (12.4%), other race (10.8%), and White (6.0%) in the young adult group (*p* < 0.001). In comparison, the edentulous older adult group was in the order of other race (34.4%), Black (26.2%), White (20.4%), Hispanic (14.8%), and Asian (11.8%), with statistically significant differences (*p* < 0.001).

Socioeconomic factors such as education and monthly income showed that rates of young adults with a full set of teeth were higher among those with low socioeconomic status (education level: below high school 14.8% vs. above high school 10.0%, *p* < 0.001; income level: below $2099: 11.9% vs. above $8400: 8.0%, *p* < 0.001). However, among older adults, there were more edentulous subjects among those of low socioeconomic status (education level: below high school 31.1% vs. above high school 12.3%, *p* < 0.001; income level: below $2099: 28.0% vs. above $8400: 5.9%, *p* < 0.001). Related to the health risk condition and healthcare access, those who had an unhealthy health condition and low healthcare access exhibited poor oral health, regardless of age group differences. Individuals with diabetes and a habit of smoking reported to be 23.7% and 67.7%, respectively, with more edentulous subjects among older adults (*p* < 0.001). Moreover, those without private insurance (23.4%) and who had last visited the dentist more than 6 months ago (31.5%) had a higher prevalence of edentulousness among older adults (*p* < 0.001).

Additionally, the prevalence of complete tooth loss among older adults was significantly higher among those with low education levels, low income, with diabetes, and those who were currently smoking. Rates of complete tooth loss was also higher among those without private insurance and who visited the dentist less often; however, there was no significant difference observed in the prevalence of complete tooth loss between men and women. However, for the younger age group (20–44 years), there were substantially different associations observed between no tooth loss and sociodemographic factors. For instance, among the younger age group, the prevalence of no missing teeth was higher than those who were Black (16.3%) compared with White (6.0%). Similarly, the prevalence of no tooth loss was 14.8% among the group with less than a high-school education (14.8%) compared with the group with more than a high-school education (10.0%).

The results of multivariable linear regression analysis to predict the mean of missing tooth (continuous measure) and the simultaneous association of predictors and outcome variables are shown in Table 3. The first model for tooth loss was statistically significant (R^2^ = 0.264; *p* < 0.001), and all other models are shown to be statistically significant. The final model for eight predictors of tooth loss was significant (R^2^ = 0.352; *p* < 0.001). Model 1 shows that the older age variable had the most substantial effect on tooth loss (≥65: β = 11.60, *p* < 0.001), and being Black (β = 1.05, *p* < 0.001) was positively associated with tooth loss, while being Hispanic and Asian was negatively associated with tooth loss. In Model 2, the effect of ethnicity on tooth loss was changed with the inclusion of the economic variables, as Hispanics had less tooth loss, while Blacks and Asians had more tooth loss, but it remained almost unchanged for Blacks after the inclusion of health risks, clinical dental risks, and health care access. After adjusting for predictors, the effect of compounding variables on tooth loss are not consistent within racial/ethnic groups, as Asians had more tooth loss from Model 1 (β = −1.97, *p* < 0.001) to Model 5 (β = −1.170, *p* < 0.001), while Blacks (model 1: β = 1.05, *p* < 0.001 to model 5: β = 0.24, *p* = 0.128) and Hispanics (Model 1 β = −0.86, *p* < 0.001 to model 5 β = −1.99, *p* < 0.001) had less tooth loss. In the full model (Model 5), older age, low education, and current smoking status are the three most important determinants of tooth loss, followed by ethnicity, low income, history of bone loss, and no recent dentist visits. Being male, Hispanic, and Asian, and having a history of gingival disease negatively predicted missing teeth. In addition, multivariate analyses with interactions on age group and income are presented in Appendix A. Related to the age group, it showed the differences in oral health by ethnicity group was a similar trend in the young adult population, while it showed different trends in the older population. As per the income, regardless of ethnicity, individuals had poor oral health when they also had a lower income level.

## 4. Discussion

The present study provided the most detailed descriptions of oral health status measured with tooth loss by age and race/ethnicity among adult populations in the U.S. Eliminating health disparities is the primary goal of Healthy People 2020, and the availability of reliable data is imperative to move toward health equity and achieve this goal. However, in Healthy People 2020, on the CDC website [23], it states, “*OH-4.2: Adults with complete tooth loss (percent, 65–74 years)*”, and there are no specific objective data for Asian and Hispanic. Thence, the findings of this study will fill the data disparities gap to provide the first report on the prevalence of complete tooth loss among Asian (11.8%) and Hispanic (14.8%) ethnic minorities based on recent population data from the National Health and Nutrition Examination Survey, from 2011 to 2019. Our findings reveal that, in general, over the last two decades, edentulism declined from 12% to 7.6% for all groups (20 and older), especially among Whites from 18% to 9.7%; however, there were no changes observed among Blacks (10% to 9.1%) and Hispanics (5% to 4.2%) from 1988–1994 NHANES [31] to 2011–2018 NHANES in the U.S. Our findings on the frequency of complete tooth loss among older Black and White adults (26.2% and 20.4%) were higher than the CDC Healthy People 2020 report (21.1% and 10.8%) [23], and this finding is counterintuitive, as the age range in our study was 65 and higher, while CDC included only those aged between 65 and 74 years. The study demonstrated that women had higher rates of tooth loss and lower numbers of full sets of teeth than men, and this result is in line with studies that have observed that women present with higher rates of tooth loss [32,33].

Our findings on older adults are in line with a previous report that a higher prevalence of complete tooth loss was found among those who were Black, with low education levels and low incomes compared with the reference groups [20,32,34]. However, to the contrary, for the young adult group, low income and low education levels showed better oral health overall, with no missing teeth. This difference can be explained by that fact that the youngest (20–44) and oldest groups (≥65) were exposed to very different life experiences than the current generation of the older adult population, and that the impact of socioeconomic factors accumulated to impact their health, including oral health. The data reported here are cross-sectional, so we can only imply the different effects of age, but cannot provide strong evidence for causation, and it cannot be ruled out that our findings are partially influenced by unobserved confounders. Hence, further studies are necessary to investigate the impact of inequality of SES and other predictors on oral health in different age groups.

Findings point out the fact that though sex differences in tooth loss are not significant in Model 1, it becomes significant for women when other predictors are taken into account. Studies report that men visit dentists less frequently compared to women, and that women are more likely to adhere to recommended treatment, even though women report more financial barriers than men [32,35,36,37]. This phenomenon of frequent dental visits and better adherence could contribute to the paradox of more dental extractions among women than men.

Our study demonstrates that oral health disparities are persistent across age groups and racial/ethnic groups, especially older adult populations in the U.S. Racial/ethnic differences remained even after controlling for other predictors. Being Hispanic and Asian was strongly negatively associated with tooth loss (β = −2.03 vs. β = −1.20; *p* < 0.001) after controlling for all other predictors. However, there was little variation in the number of missing teeth for multivariate adjusted analyses among Blacks, as the relationship between being Black and tooth loss was not significant once SES, physical risk factors, clinical dental condition, and health care access factors were included. On the contrary, after adding models with more predictors, Asians showed more tooth loss (β = −1.98 to β = −1.20), while Hispanics had less tooth loss (β = −0.86 to β = −2.30) from Models 1 to 5. Our data suggest that the proposed covaries that are commonly used in epidemiology studies may not capture race-/ethnic group-specific indicators for oral health, especially related to missing teeth among diverse older adult populations. Additionally, it is possible that better oral health among Asian and Hispanic minorities in the U.S. could be attributable to the healthy migrant phenomenon [38]. The findings show that Hispanics are one ethnicity with lower tooth loss, but this does not change with the inclusion of all covaries, while the Asian population demonstrated the highest changes in tooth loss. Further studies are a necessity to explore the different directions of the influence of covaries or the selection bias of covaries and the sample.

### Strengths and Limitations

This study has several strengths and limitations as well. The most important strength is a large sample size, which provides the power to evaluate the effect of covaries. The second is the NHANES survey methods. The inter-reliability of the oral health information, particularly the tooth count, was high between dental examiners (Kappa score: 0.96–1.00) [27]. Third, a multistage probability sampling design allowed NHANES to oversample racial/ethnic minorities and gain representative data about them. NHANES have oversampled Asian Americans since 2011 [39]. For data collection from participants with low English proficiency, NHANES uses nationwide translation interpreter services when language barriers are detected during eligibility screening. Consequently, this study showed that Asian Americans were successfully oversampled when compared to the census (12.3% vs. 6%) [40]. Lastly, the study cumulatively contains eight years of national survey data that tried to achieve external validity.

However, this study also has limitations. Although NHANES used diverse strategies to gain representative data from minorities, there still is a gap that may discourage minorities’ participation or gain data that may not represent minorities with low English proficiency. For example, an advance letter that NHANES sent to selected households was prepared only in English and Spanish. Moreover, the NHANES questionnaire was translated only into English and Spanish, which means that the data quality from non-English speaking participants may have varied based on the interpreters who accompanied the data collectors. Additionally, the study design was a cross-sectional study, and there is no evidence to show causation from the results. The sociodemographic, health conditions and behaviors, healthcare access, and oral health (except for tooth loss) data were collected in a self-report format that could have had recall bias. This study had the number of missing teeth as its outcome, and missing wisdom teeth were also factored in as the definition of missing teeth in this study, applied conservatively. Although the study adjusted for confounding variables, there are potential confounding variables that are not included in this study; for example, among the study population, some missing teeth were wisdom teeth that were less likely to be related to poor oral health. In addition, due to the characteristics of NHANES, the study was not able to include clinical data for the adjustment.

## 5. Conclusions

Our study provided oral health outcomes, measured with tooth loss for four racial/ethnic adult groups based on a nationally representative sample among adults [28]. As expected, the complete tooth loss was significantly high among older adults, that points to the fact that subgroup-tailored prevention, such as providing dentures and screening cognitive impairment for older adults, needs to be considered. Though racial/ethnicity differences remained even after controlling for all other predictors, the impact of predictors on each racial/ethnic group is varied when SES, physical, clinical dental, health behavior, and healthcare access factors are included in the model. Thus, its interpretation of the effect of the predictors of tooth loss on racial/ethnicity requires a caution, as it is difficult to drive causality and temporal issues from cross-sectional analysis. Longitudinal studies could be helpful to establish causal relationships. In addition, this study reveals several differences between men and women and different ethnic groups related to oral health, and highlights the need for further study to better understand these differences and develop prevention and education tailored to specific age, sex, and ethnicity to improve oral health among adults.

## Figures and Tables

**Table 1 ijerph-19-02382-t001:** Selected sociodemographic characteristics and risk factors for tooth loss in adults, NHANES, 2011–2018 (*n* = 15,161).

	Total		Missing Tooth							
	N	%	None (0)	1~5	6~31	All (32)	*p*-Value ^†^	Mean	S.E	*p*-Value ^‡^
			%	%	%	%				
**Total**	15,161		5.8	45.2	41.4	7.6	<0.001 ^“^	9.1	9.0	<0.001 ^“^
**Age group**							<0.001			<0.001
20~44	5723	37.7	10.9	68.3	20.1	0.7		4.4	4.1	
45~64	5688	37.5	3.9	39.4	50.5	6.2		9.4	8.4	
≥65	3750	24.7	1.0	18.6	60.2	20.2		16.1	10.7	
**Sex**							<0.001			0.666
Male	7386	48.7	61.8	48.7	46.5	51.1		9.1	9.3	
Female	7775	51.3	38.2	51.3	53.5	48.9		9.2	8.8	
**Race**							<0.001			<0.001
Hispanic	3468	22.9	7.2	47.2	41.4	4.2		8.0	8.0	
Non-Hispanic White	5980	39.4	3.2	45.4	41.6	9.7		9.9	9.5	
Non-Hispanic Black	3300	21.8	7.4	37.2	46.4	9.1		10.3	9.7	
Non-Hispanic Asian	1867	12.3	8.8	54.8	33.4	3.1		6.5	6.9	
Other Race—Including Multi-Racial	546	3.6	6.8	44.7	37.5	11.0		9.7	9.8	
**Education level**							<0.001			<0.001
Below high school	3229	21.3	6.0	31.7	47.6	14.6		12.5	10.9	
Graduated high school	3385	22.3	4.9	38.6	46.9	9.6		10.5	9.7	
Above high school	8547	56.4	6.1	52.8	37.0	4.1		7.4	7.4	
**Income (month)**							<0.001			<0.001
$0~$2099	5490	36.2	5.1	35.0	47.2	12.7		11.7	10.4	
$2100~$5399	5713	37.7	6.1	45.2	42.4	6.3		8.8	8.6	
$5400~$5399	2008	13.2	6.4	55.3	35.4	2.8		6.7	6.7	
$8400~	1950	12.9	6.4	63.4	28.7	1.6		5.6	5.1	
**Born in US**							<0.001			<0.001
Yes	8521	56.2	5.8	54.7	53.1	11.0		9.8	9.3	
No	3783	25.0	10.4	57.8	46.5	5.5		7.7	8.0	
**Health insurance**							<0.001			<0.001
Yes	12,509	82.5	5.1	44.3	42.2	8.3		9.5	9.2	
No	2652	17.5	9.1	49.2	37.7	4.0		7.3	7.7	
**Private insurance**							<0.001			<0.001
Yes	7786	51.4	5.9	51.1	37.9	5.1		7.9	7.9	
No	7375	48.6	5.8	38.9	45.1	10.2		10.5	9.9	
**Diabetes**							<0.001			<0.001
Yes	2218	14.6	2.8	26.5	55.5	15.2		13.6	10.5	
No	12,943	85.4	6.4	48.4	39.0	6.3		9.4	8.5	
**Current smoking status**							<0.001			<0.001
Every day	1756	11.6	4.2	33.0	49.5	13.2		12.1	10.4	
Some days	410	2.7	7.3	46.8	39.5	6.3		8.4	8.6	
Not at all	12,995	85.7	6.0	46.7	40.4	6.8		8.8	8.8	
**Last visit to dentist**							<0.001			<0.001
6 months or less	5538	36.5	3.9	48.1	45.6	2.4		7.8	6.9	
More than 6 months	9623	63.5	7.0	43.5	39.0	10.5		9.9	10.0	
**History of bone loss (oral)**							<0.001			<0.001
Yes	2092	13.8	2.1	26.4	61.2	10.4		12.2	9.5	
No	13,069	86.2	6.4	48.2	38.3	7.1		8.7	8.8	
**History of gum disease**							<0.001			0.001
Yes	3377	22.3	4.1	41.1	51.2	3.6		8.7	7.6	
No	11,784	77.7	6.3	46.3	38.6	8.7		9.3	9.4	
**Self-rated gum health**							<0.001			<0.001
Very good	5745	37.9	7.9	58.5	25.5	8.0		7.4	9.5	
Good	4693	31.0	5.4	42.9	42.1	9.6		9.7	8.6	
Poor	4723	31.2	3.7	31.1	60.1	5.0		10.7	8.7	

^†^ Results from the Chi-square test between variables. ^‡^ Results from *t*-test or ANOVA between variables. ^“^ Result from univariate analysis on missing teeth. Bold shows its category.

**Table 2 ijerph-19-02382-t002:** Prevalence of tooth loss by sociodemographic characteristics and risk factors by age (*n* = 15,161).

	20–44 years, Number of Missing Teeth (*n* = 5723)					45–64 years, Number of Missing Teeth (*n* = 5688)					≥65 years, Number of Missing Teeth(*n* = 3750)				
	Total	None (0)	1~5	6~31	All (32)	Total	None (0)	1~5	6~31	All (32)	Total	None (0)	1~5	6~31	All (32)
		%	%	%	%		%	%	%	%		%	%	%	%
	5723	10.9	68.3	20.1	0.7	5688	3.9	39.4	50.5	6.2	3750	1.0	18.6	60.2	20.2
**Sex**															
Male	2721	14.4	67.5	17.2	0.8	2776	4.8	41.1	47.9	6.2	1889	1.2	18.7	59.4	20.7
Female	3002	7.8	69.0	22.7	0.6	2912	3.1	37.8	53.1	6.1	1861	0.9	18.5	61.0	19.7
**Race**															
Hispanic	1409	12.9	69.0	17.9	0.2	1368	4.2	41.0	51.8	3.1	691	1.3	15.1	68.9	14.8
Non-Hispanic White	2060	6.0	69.5	23.1	1.5	2012	2.7	42.3	46.8	8.1	1908	0.8	22.7	56.1	20.4
Non-Hispanic Black	1179	16.3	62.8	20.7	0.3	1370	3.2	29.8	59.6	7.4	751	0.9	10.4	62.5	26.2
Non-Hispanic Asian	806	12.4	72.2	15.3	0.1	757	7.5	48.1	41.7	2.6	304	2.3	25.3	60.5	11.8
Other Race—Including Multi-Racial	269	10.8	67.7	20.4	1.1	181	4.4	31.5	50.8	13.3	96	0.0	5.2	60.4	34.4
**Education level**															
Below high school	946	14.8	58.0	25.5	1.7	1235	3.7	30.5	55.1	10.6	1048	0.9	9.4	58.6	31.1
Graduated high school	1221	10.7	64.9	23.6	0.8	1288	2.3	31.4	58.0	8.3	876	0.6	12.8	63.0	23.6
Above high school	3556	10.0	72.2	17.5	0.4	3165	4.6	46.2	45.7	3.5	1826	1.3	26.6	59.8	12.3
**Income (month)**															
$0~$2099	1813	11.9	63.6	22.9	1.6	2042	2.6	28.2	58.7	10.4	1635	0.7	11.6	59.7	28.0
$2100~$5399	2267	11.7	66.5	21.4	0.4	1996	3.8	39.0	52.1	5.2	1450	0.8	20.3	61.9	17.0
$5400~$5399	871	9.6	73.4	16.8	0.2	758	4.6	47.8	45.1	2.5	379	2.6	29.0	58.8	9.5
$8400~	772	8.0	78.8	13.2	0.0	892	6.4	59.0	33.1	1.6	286	1.7	35.7	56.6	5.9
**Private Insurance**															
Yes	2976	9.9	72.3	17.6	0.2	3043	4.8	47.4	44.6	3.3	1767	1.1	21.6	60.7	16.6
No	2747	12.1	63.9	22.8	1.3	2645	2.8	30.3	57.4	9.5	1983	0.9	15.9	59.8	23.4
**Diabetes**															
Yes	239	8.4	60.7	29.3	1.7	974	3.4	30.1	56.7	9.9	1005	0.9	14.9	60.5	23.7
No	5484	11.1	68.6	19.7	0.7	4714	4.0	41.3	49.3	5.4	2745	1.1	19.9	60.1	18.9
**Current smoking status**															
Every day	704	9.5	55.7	31.3	3.6	804	0.7	21.0	64.2	14.1	248	0.4	7.7	54.0	37.9
Some days	221	13.1	64.7	21.7	0.5	142	0.7	31.7	59.9	7.7	47	0.0	8.5	61.7	29.8
Not at all	4798	11.0	70.3	18.4	0.3	4742	4.5	42.8	48.0	4.8	3455	1.1	19.5	60.6	18.8
**Last visit to dentist**															
6 months or less	1602	6.1	68.2	25.0	0.7	2358	4.1	47.2	46.5	2.1	1578	1.3	28.9	65.2	4.6
More than 6 months	4121	12.8	68.3	18.2	0.7	3330	3.7	33.9	53.4	9.0	2172	0.8	11.1	56.6	31.5
**History of bone loss (oral)**															
Yes	343	6.7	49.3	39.9	4.1	1005	1.8	26.5	63.5	8.3	744	0.3	15.7	67.9	16.1
No	5380	11.2	69.5	18.8	0.5	4683	4.3	42.2	47.8	5.7	3006	1.2	19.3	58.3	21.2
**History of gum disease**															
Yes	945	7.9	64.9	26.5	0.7	1545	3.6	38.3	55.2	3.0	887	0.9	20.7	70.6	7.8
No	4778	11.5	68.9	18.8	0.7	4143	4.0	39.9	48.8	7.3	2863	1.0	17.9	57.0	24.0

Univariate analysis was used to test the differences of distribution of tooth loss within the age group. The Chi-square test was to test the difference in distribution of tooth loss by gender, race, socioeconomic status, healthcare access, and health conditions and health behavior. All differences were significant (*p* < 0.001), except for sex in the ≥65-year age group (*p* = 0.627) and health insurance in the ≥65-year age group (*p* = 0.474). Bold shows its category.

**Table 3 ijerph-19-02382-t003:** Stepwise regression analysis of tooth loss for sociodemographic characteristics and risk factors among adults in the U.S., NHANES, 2011–2018 (*n* = 15,161).

	Model 1			Model 2			Model 3			Model 4			Model 5		
	β	S.E.	*p*-Value	β	S.E.	*p*-Value	β	S.E.	*p*-Value	β	S.E.	*p*-Value	β	S.E.	*p*-Value
**Age group**															
20~44	Ref.			Ref.			Ref.			Ref.			Ref.		
45~64	5.01	0.15	<0.001	4.77	0.14	<0.001	4.50	0.14	<0.001	4.40	0.14	<0.001	4.69	0.14	<0.001
≥65	11.60	0.16	<0.001	10.75	0.16	<0.001	10.61	0.16	<0.001	10.41	0.17	<0.001	10.77	0.17	<0.001
**Sex**															
Male	−0.28	0.13	0.027	−0.35	0.12	0.004	−0.50	0.12	<0.001	−0.48	0.12	<0.001	−0.60	0.12	<0.001
Female	Ref.			Ref.			Ref.			Ref.			Ref.		
**Race**															
Hispanic	−0.86	0.17	<0.001	−2.18	0.17	<0.001	−1.94	0.17	<0.001	−1.82	0.17	<0.001	−1.99	0.17	<0.001
Non-Hispanic White	Ref.			Ref.			Ref.			Ref.			Ref.		
Non-Hispanic Black	1.05	0.17	<0.001	0.45	0.16	0.006	0.42	0.16	0.010	0.47	0.16	0.003	0.24	0.16	0.128
Non-Hispanic Asian	−1.98	0.21	<0.001	−1.43	0.20	<0.001	−1.21	0.20	<0.001	−1.09	0.20	<0.001	−1.17	0.20	<0.001
Other Race—Including Multi-Racial	1.49	0.35	<0.001	1.23	0.34	<0.001	1.17	0.33	<0.001	1.15	0.33	0.001	0.99	0.33	0.002
**Education level**															
Below high school				3.46	0.17	<0.001	3.20	0.17	<0.001	3.22	0.17	<0.001	2.91	0.17	<0.001
Graduated high school				1.94	0.16	<0.001	1.78	0.16	<0.001	1.77	0.15	<0.001	1.59	0.15	<0.001
Above high school				Ref.			Ref.			Ref.			Ref.		
**Income (month)**															
$0~$2099				3.53	0.21	<0.001	3.06	0.21	<0.001	3.03	0.21	<0.001	2.29	0.22	<0.001
$2100~$5399				1.85	0.20	<0.001	1.61	0.20	<0.001	1.59	0.20	<0.001	1.12	0.20	<0.001
$5400~$5399				0.91	0.24	<0.001	0.80	0.24	0.001	0.81	0.23	0.001	0.63	0.23	0.007
$8400~				Ref.			Ref.			Ref.			Ref.		
**Diabetes**															
Yes							1.89	0.18	<0.001	1.85	0.18	<0.001	1.76	0.17	<0.001
No							Ref.			Ref.			Ref.		
**Current smoking status**															
Yes							2.97	0.19	<0.001	2.87	0.19	<0.001	2.69	0.19	<0.001
No							Ref.			Ref.			Ref.		
**History of bone loss (oral)**															
Yes										2.28	0.19	<0.001	2.50	0.15	<0.001
No										Ref.			Ref.		
**History of gum disease**															
Yes										−1.47	0.15	<0.001	−1.23	0.15	<0.001
No										Ref.			Ref.		
**Private insurance**															
Yes													Ref.		
No													0.28	0.13	0.039
**Last visit to dentist**															
6 months or less													Ref.		
More than 6 months													2.20	0.13	<0.001
**R^2^**	0.264			0.316			0.332			0.340			0.352		

Bold shows its category.

## Data Availability

Data are available from National Centers for Health Statistics under the Centers for Disease Control and Prevention (https://www.cdc.gov/nchs/nhanes/index.htm, accessed date: 9 February 2022) for public use.

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
