# Peer review of "Ethnicity, Social, and Clinical Risk Factors to Tooth Loss among Older Adults in the U.S., NHANES 2011–2018"

_ijerph, 2022, doi:10.3390/ijerph19042382_

Round 1

Reviewer 1 Report

The paper is well written and well discussed. It is accepted in present form.

Author Response

Thank you so very much for your comments as well as your encouraging words

Reviewer 2 Report

I have read the manuscript "Ethnicity, Social, and Clinical Risk Factors to Tooth Loss among Older Adults in the U.S., NHANES 2011-2018" with great interest. The authors gave detailed descriptions of oral health status measured with tooth loss by age and race/ethnicity among adult populations in the U.S. Although the topic is partly covered and cleared before, in this study they presented new data for elder Asian and Hispanic Populations on complete tooth loss. They also discussed other covariates including, demographic (gender, age, and race/ethnicity), socioeconomic (education and income), health risk(oral health related medical conditions), oral health (periodontal bone loss), and health care access (health care insurance and dental care visits) domains. 

This is a well-written manuscript, I have minor recommendations:

  1. I wonder if there could be other covariates in the health risk domain other than diabetes and smoking that are considered to be related to oral health. It is known that many other health conditions even the gut microbiome change are considered to be related to oral health. Are there any other health risk covariates that authors can include from the NHANES database? If so how did they decide on only diabetes and smoking? 
  2. It would be nice if authors should provide references for the covariates that they have chosen and their possible relation to oral health so it is clear for everyone why all those parameters are chosen. I can understand that they have used standard covariates that we mostly use in epidemiological studies but the selection bias (both sample and covariates) is a big issue in large those kinds of large-scale analysis.
  3. I am a bit confused about the statistical significance. The authors mention that they accepted statistical significance as <0.01 but then under table 2 they wrote the test for diabetes was not significant p=0.001  which should be significant in this case? ANd also when they present results they stated mostly p<0.0001 which can be but if they accepted 0.01 as significance they should just say <0.01 how small the significance is good to know but confusing. They then should just write p<0.01 - I would recommend authors to check the whole manuscript according to this inconsistency.

Author Response

Dear Reviewer 2,

Thank you so very much for your comments as well as the outstanding suggestions and comments made by the reviewers.

We have diligently reviewed each comment and have addressed the concerns and suggestions made. Below you will find detailed responses to each comment.

We are excited about the potential to have this manuscript published by the International Journal of Environmental Research and Public Health. We welcome any further suggestions and/or comments.

Sincerely,

All Authors

Reviewer 3 Report

While the manuscript presents, in my view, some issues on the interpretation of the results, the findings on the Asian populatian may provide important insights to researchers without taking the importance of the findings for the other race.

This manuscript is also quite attractive to understand the factors behind tooth loss  other than ethnicity, in the U.S.

 I have the following comments for the authors' consideration.

Abstract: 

✓ Background- In this study cognitive impairment and dementia in older adults seems 
inadequate. The socioeconomical burden of oral disease would be more important. 
✓ Methods: It is difficult to understand the abstract without first providing (brief) context as 
to “adjusting for predictors” and concurrently, the reference to Model 1 to 5 (?) 
✓ Conclusion could be improved 
o I find poor “ As expected, complete tooth loss was significantly higher among 
older adults”- this is the only conclusion???? 
o Again “cognitive impairment and dentures” are not the basis of this study. 

Introduction: 

The introduction is very-well organized and divided and it was used updated references although, 
I have a few considerations: 

✓ When refering “studies provide evidence of an association between tooth loss and 
dementia, which was stronger for older adults [6,7,15]”, I think reference 7 should be 
removed, here the association is Alzheimer and Periodontitis and changed for one more 
appropriated: 
o Cerutti-Kopplin, D.; Feine, J.; Padilha, D.M.; de Souza, R.F.; Ahmadi, M.; 
Rompre, P.; Booij, L.; Emami, E. Tooth Loss Increases the Risk of Diminished 
Cognitive Function: A Systematic Review and Meta-analysis. JDR Clin. Transl. 
Res. 2016, 1, 10–19. 

✓ The poor oral conditions in the older individuals can also be a consequence of Diabetes 
– which the authors mention- needs a reference in the text. 
✓ A paragraph about nutritional status and habits in the diferente ethnicities, the use of 
dentures, and reduced masticatory function in the elderly adults sould be included. 
✓ Line 66-67- edentulism AND cognitive impairment and dementia- again, as in the Abstract 
I don`t think it should be “used” as the basis of the study 

Materials and methods this section is very-well described although it could be improved: 

✓ It is importante to describe the areas searched- rural versus urban- it is known that it can 
change the outcomes. 
✓ The 8 predictors are unclear 
✓ In the correlated variables the 2 domains – health risk and oral health status ( the risk- 
why just Smoke and not alcoohl should be explained) and why just Diabetes when there 
are other systemic diseases corelated. 
✓ In the data analysis . 
o Line 134- The first model was adjusted for age, sex, and ethnicity - interchanges 
gender /sex- it should be the same all over the text 

Findings: I woud say Results 

The tables are demonstrating the results and seem adequate to the study performed 

✓ Correct the first sentence: 

Line 147- “From the study sample ………23% were Hispanic, 39%were non-
Hispanic White, 22% non-Hispanic Black, and 12% non-Hispanic Asian. 

✓ Line 166- The prevalence of complete tooth loss among older adults 166 (>65 years) was 
significantly higher among Black (26.2%) compared with White (20.4%), 167 Hispanic 
(14.8%) and Asian (11.8%). 

This is already in Table 2 and not Table 1( wrongly referenced) 

✓ Line 198-202- Repeated full sentence – cut ( the same in line 168-72) 

Discussion 

✓ Line 249 “ The study demonstrated that women had higher tooth loss and lower numbers 
of full sets of teeth than men”- It´s not supported by the results showed…..if it is- 
numbers?! 
o Tooth loss should be well separated from dental extractions 

✓ Line 254- “However, to the contrary, for the young adult group, low income and low 
education levels showed better oral health measured with no missing tooth” 
o This result could be further explained 

✓ Line 281- “Our data suggest that the proposed predictors that were commonly used 
determinants for other health indicators may not capture race/ethnic group specific 
indicators for oral health, especially related to missing teeth among adult populations.” 
o Examples of other predictors????? 

✓ The last sentences could be further explored- the Hispanics are one ethnicity with the 
lower % of tooth loss- this does`t change with the inclusion of all predictors. What was 
added in the results was the Asian population that also had one of the highest % of tooth 
loss. 

✓ With the inclusion of the Asian ethcniicity it was said in the Abstract that “predictors impact 
in tooth loss was lost/not consistent.” This sentence removes all the importance of the 
previous results of this group and others like: 
o Wu B, Liang J, Plassman BL, Remle RC, Bai L. Oral health among white, black, 
and Mexican-American elders: an examination of edentulism and dental caries. 
J Public Health Dent. 2011 Fall;71(4):308-17. 
o Nazer FW, Sabbah W. Do Socioeconomic Conditions Explain Ethnic Inequalities 
in Tooth Loss among US Adults? Ethn Dis. 2018 Jul 12;28(3):201-206. 

✓ Regarding Asian population in the U.S. it should be further studied, specially if there is 
no data from them in NHANES as said in the introduction. This scientific discussion 
should/could be enriched 
o South Asian populations represent a big proportion of the low-income migrants 
in the US and the children have one of the highest rates of caries among all 
ethnic/racial groups- is this related with cultures, health care system, nutrition 
diferences? 

Author Response

Dear Reviewer 3,

Thank you so very much for your comments as well as the outstanding suggestions and comments made by the reviewers.

We have diligently reviewed each comment and have addressed the concerns and suggestions made. Below you will find detailed responses to each comment.

We are excited about the potential to have this manuscript published by the International Journal of Environmental Research and Public Health. We welcome any further suggestions and/or comments.

Sincerely,

All Authors 

Reviewer 4 Report

The objective of this study was to investigate the determinants of tooth loss in a NHANES study (USA).

I have some concerns about the data presented by the authors :

  • In line 99 the authors define tooth loss as partial or complete. However, in  Table 1, it has been classified into 0, 1~5, 6~31, All. Please update Methods accordingly.
  • In Table 1 the statistical test performed is not clear (what is compared). Please add in the legend of the table what the tested comparison represents using the statistical tests performed.
  • The authors reported more prevalence of tooth loss among women. While counterintuitive, this has been justified in discussion. Another explanation would be hormones (Meisel, 33 cited by the authors). This demonstrates the need to examine the role of interactions in multivariate regression. For example how is the interaction age group*gender ?
  • Since the authors put a particular focus on ethnicity, it would be necessary to explore some of the other interactions such as income with ethnicity in the multivariate model.
  • Line 66-67 : "Edentulism has been associated with cognitive impairment and dementia [24-26] and 66 it is imperative to provide baseline data of tooth loss and its correlators reflecting race/ethnicity among adult populations.". Why talking about dementia whereas there is no data about it nowhere else in the manuscript?

Author Response

Dear Reviewer 4,

Thank you so very much for your comments as well as the outstanding suggestions and comments made by the reviewers.

We have diligently reviewed each comment and have addressed the concerns and suggestions made. Below you will find detailed responses to each comment.

We are excited about the potential to have this manuscript published by the International Journal of Environmental Research and Public Health. We welcome any further suggestions and/or comments.

Sincerely,

All Authors 

Reviewer 5 Report

The aim of the present study is to examine ethnic differences in tooth loss among American young, middle and older adults and, to evaluate the effect of bio-socio-culture-health behavior-health factors on tooth loss.

You can see below some suggestions to improve the manuscript:

  • In the introduction section the Authors state: “Studies provide evidence of an association between tooth loss and dementia, which was stronger for older adults” ;moreover the Authors state: “Edentulism has been associated with cognitive impairment and dementia and it is imperative to provide baseline data of tooth loss and its correlators reflecting race/ethnicity among adult populations”.  This aspect is not further investigated in the study and is not an aim of the study. I suggest to delate it from the text or to reduce the sentences and references about dementia.
  • The Authors collect data of the Oral Health domain that included periodontal bone loss and previous gingival disease. How these data are collected? How is evaluated the bone loss? A simple survey is not enough to make a quantitative assessment of bone loss and gingival diseases. I think it should be added as a limitation of the study;
  • There are some double spaces in the text. Please do a general check of te text;
  • There are errors in the references. Please correct.

Author Response

Dear Reviewer 5,

Thank you so very much for your comments as well as the outstanding suggestions and comments made by the reviewers.

We have diligently reviewed each comment and have addressed the concerns and suggestions made. Below you will find detailed responses to each comment.

We are excited about the potential to have this manuscript published by the International Journal of Environmental Research and Public Health. We welcome any further suggestions and/or comments.

Sincerely,

All Authors 
